# Poor correlation between large-scale environmental flow violations and freshwater biodiversity: implications for water resource management and the freshwater planetary boundary

Chinchu Mohan[1,2,3*], Tom Gleeson[2,4*], James S Famiglietti[1,5], Vili Virkki[6], Matti Kummu[6], Miina Porkka[6,7], Lan Wang-Erlandsson[8,9], Xander Huggins[1,2], Dieter Gerten[10,11], Sonja C. Jähnig[11,12]

[1] Global Institute for Water Security, University of Saskatchewan, Saskatoon, Saskatchewan, Canada.

[2] Department of Civil Engineering, University of Victoria, Victoria, British Columbia, Canada

[3] Waterplan (YC S21), San Francisco, California, USA

[4] School of Earth and Ocean Sciences, University of Victoria, Victoria, British Columbia, Canada

[5] School of Environment and Sustainability, University of Saskatchewan, Saskatoon, Saskatchewan, Canada.

[6] Water and Development Research Group, Aalto University, Espoo, Finland

[7] Global Economic Dynamics and the Biosphere, Royal Swedish Academy of Sciences, Stockholm, Sweden

[8] Stockholm Resilience Centre, Stockholm University, Stockholm, Sweden

[9] Bolin Centre for Climate Research, Stockholm University, Stockholm, Sweden

[10] Potsdam Institute for Climate Impact Research (PIK), Member of the Leibniz Association, Potsdam, Germany

[11] Humboldt-Universität zu Berlin, Geography Department and Integrative Research Institute on Transformations of Human–Environment Systems, Berlin, Germany

[12] Leibniz Institute of Freshwater Ecology and Inland Fisheries, Müggelseedamm 310, Berlin, Germany

*Correspondence to*: Chinchu Mohan (chinchu.mohan@usask.ca), Tom Gleeson (tgleeson@uvic.ca)

ORCID corresponding authors: Chinchu Mohan: 0000-0001-7611-3392; Tom Gleeson: 0000-0001-9493-7707

**Key Research Points**

- No significant relationship between environmental flow (EF) violation and freshwater biodiversity indicators was found at global or ecoregion scales using globally consistent methods and currently available data, when not accounting for other factors affecting freshwater biodiversity.

- Several basins show a slight positive correlation between EF violation and biodiversity indicators, which could be attributed to the artificial introduction of non-native species.

- A generalized approach that incorporates EF considerations but ignores the lack of a significant EF-biodiversity relationship at large scales can underestimate the stress on the

31        ecosystem at smaller scales which correspond with eco-hydrological processes that

32        determine ecological impacts from EF violation.

●   Use of a globally aggregated blue water planetary boundary using biodiversity-based

34        response variables is deceptive

## Abstract

The freshwater ecosystems around the world are degrading, such that maintaining
environmental flow[1] (EF) in river networks is critical to their preservation. The relationship
between streamflow alterations and, respectively, EF violations[2], and freshwater biodiversity is
well established at the scale of stream reaches or small basins (~<100 km²). However, it is unclear
if this relationship is robust at larger scales even though there are large-scale initiatives to legalize
the EF requirement. Moreover, EFs have been used in assessing a planetary boundary[3] for
freshwater. Therefore, this study intends to conduct an exploratory evaluation of the relationship
between EF violation and freshwater biodiversity at globally aggregated scales and for freshwater
ecoregions. Four EF violation indices (severity, frequency, probability to shift to violated state,
and probability to stay violated) and seven independent freshwater biodiversity indicators
(calculated from observed biota data) were used for correlation analysis. No statistically
significant negative relationship between EF violation and freshwater biodiversity was found at
global or ecoregion scales. These findings imply the need for having a holistic bio-geo-hydro-
physical approach in determining the environmental flows. While our results thus suggest that
streamflow and EF may not be an only determinant of freshwater biodiversity at large scales,
they do not preclude the existence of relationships at smaller scales or with more holistic EF

---

[1] Environmental flow (EF): "The quantity, timing, and quality of water flows required to sustain freshwater and estuarine ecosystems and the human livelihoods and well-being that depend on these ecosystems." - Arthington et al., 2018

[2] EF violations: EF violations are deviations in streamflow beyond the upper and lower boundary of Environmental Flow envelopes (EFE). The EFE establish an envelope for acceptable EF deviations based on pre-industrial (1801-1860) stream discharge (See section 2.2 for more details)

[3] Planetary boundary: Planetary boundary defines biogeophysical planetary scale boundaries for Earth system processes that, if violated, can irretrievably impair the Holocene-like stability of Earth system (see box 1 for more details)

methods (e.g., including water temperature, water quality, intermittency, connectivity etc.) or
with other biodiversity data or metrics.

**Keywords**: Environmental flow violation, freshwater biodiversity, Global scale, freshwater
ecoregions.

## 1.Introduction

Water resources are inarguably one of the most important natural resources in the Earth system
for sustaining life. Nevertheless, these resources and their associated ecosystems are threatened
by human actions (Bélanger and Pilling, 2019; Clausen and York, 2008; Vörösmarty et al., 2010;
Wilting et al., 2017). Global freshwater covers up to 0.8% of the total Earth's surface (Gleick,
1996) and inhabits 6% of all the known species in the world including 40% of total fish diversity
and nearly one third of all vertebrates (Lundberg et al., 2000). Since freshwater ecosystems have
high species richness in a relatively small area and are exposed to a high level of pressure, they
are more vulnerable to environmental change and human actions than any other ecosystems
(Dudgeon et al., 2006). The rapid increase in the demand for natural resources is the fundamental
cause for freshwater ecosystem degradation (Darwall et al., 2018). Anthropogenic climate
change (Allan and Flecker, 1993; Darwall and Freyhof, 2016; Knouft and Ficklin, 2017; Meyer et
al., 1999), overexploitation (Allan et al., 2005), water pollution (Albert et al., 2021; Dudgeon et
al., 2006; Reid et al., 2019; Smith, 2003), flow alteration (Nilsson et al., 2005; Vorosmarty et al.,
2000), habitat destruction (Dudgeon, 2001) and introduction of alien species (Gozlan et al., 2010;
Vitule et al., 2009) are some of the manifestations of this increased demand which directly
threatens the freshwater ecosystems. In addition, increased water impoundment in large dams
and reservoirs has also led to an array of adversities to freshwater ecosystems ranging from
habitat destruction to irregular flow alterations (Bergkamp et al., 2000). This situation is
aggravated by increasing pressure on related Earth system functions, such as climate change and
nutrient cycles, which are articulated by their respective transgressions in the planetary
boundaries framework (Box 1) (Dudgeon, 2010). Freshwater ecosystem processes that were
previously governed by natural Earth system facets such as temperature, rainfall, and relief are
now increasingly driven by demographic, social, and economic drivers (Clausen and York, 2008;
Kabat et al., 2004; Tyson et al., 2002; Vitousek et al., 1997; Vörösmarty et al., 1997). Freshwater
ecosystem health comprises both biotic factors like biodiversity and abiotic factors like habitat
integrity. As any disruption in the abiotic factors is most likely to be reflected in the biotic status
of the freshwater ecosystem, the scope of this paper is confined to the biotic dimension of the
freshwater ecosystem (i.e., biodiversity) and not the health of the entire ecosystem.

There has been an increased recognition in recent decades for the need of maintaining a natural
flow regime in streams to sustain healthy ecosystems. (Horne et al., 2017; Poff et al., 1997, 2017;
Tickner et al., 2020; Tonkin et al., 2021). Despite the indispensable role of aquatic biodiversity in
maintaining the quality of the system (Darwall et al., 2018), inclusion of such environmental flow
(EF) in water management is often controversial, particularly in regions where freshwater
availability is limited and is already a matter of severe competition. These competitions have led
to an increasing trend in EF violation (insufficient streamflow than the recommended EF
requirement; see section 2.1 for more details) in the past decade both in terms of severity and
frequency (Virkki et al., 2022). This wakeup call has led to several international and national
efforts to legalize EF requirements through large-scale EF management schemes (Arthington and
Pusey, 2003; Richter et al., 1997, 2003). The Water and Nature Initiative (Smith and Cartin, 2011),
the Brisbane declaration (Declaration, 2007), and the Global Action Agenda (Arthington et al.,
2018) are some of these efforts. Nevertheless, there is a large gap in our understanding of the
relationship between EF requirements and biodiversity responses at various spatial and temporal
scales. Except for a few (Domisch et al., 2017; Xenopoulos et al., 2005; Yoshikawa et al., 2014),
the majority of the studies exploring this relation were conducted at smaller scales (Anderson et
al., 2006; Arthington and Pusey, 2003; Powell et al., 2008). Thus, there is a significant discrepancy
in the scale at which these processes are understood versus the scale at which the policies are
set (Thompson and Lake, 2010). Current knowledge of how the small-scale processes scale up
(e.g., validation of large-scale EF hydrologic methods using local data) to a regional or global scale
is thus limited, potentially undermining the scientific integrity of existing large-scale EF
management schemes.

In order to scientifically underpin large scale EF policies, the existing assumption of the inverse
relationship between freshwater biodiversity response and EF violation must be tested at
regional and global scales (see Supplementary information S1 for more details). Therefore, in this
study, we evaluate the relationship between EF violation and freshwater biodiversity at two
different spatial scales (freshwater ecoregion, global) using four EF violation indices (frequency,
severity, probability to move to a violated state, and probability to stay violated) and seven
freshwater biodiversity indicators describing taxonomic, functional, and phylogenetic
dimensions of the biodiversity. The paper is not intended to be a definitive test on the
relationship between EF violation and aquatic biodiversity. It is rather intended to be an
exploratory analysis of the idea of conducting more detailed evaluations of the EF-biodiversity
relationship before formulating large scale EF management policies. The implications of the
findings for large-scale water management and the use of the relationship between
environmental flows and freshwater biodiversity (hereafter referred to as EF-biodiversity
relationship) in the planetary boundary framework (box 1) are also discussed.

---

**Box 1: Introduction to blue water planetary boundary framework**

The planetary boundaries framework proposed by Rockström et al. (2009) and further developed by Steffen et al. (2015) defines bio geophysical planetary scale boundaries for Earth system processes that, if violated, can irretrievably impair the Holocene-like stability of Earth system. The framework establishes scientifically determined safe operating limits for human perturbations through control and response variable relationships, under which humans and other life forms will coexist in equilibrium without jeopardizing the Earth's resilience. Nine planetary boundaries were defined to cover all independent significant Earth system processes. Out of the nine, the freshwater planetary boundary quantifies the safe limits of the terrestrial hydrosphere (Gleeson et al., 2020a, b).

---

The freshwater planetary boundary was originally defined using human water consumption as the control variable, set at 4000 km$^3$/yr (with an uncertainty of 4000 to 6000 km$^3$/yr) (Rockström et al., 2009). Gerten et al. (2013) proposed a bottom-up, spatially explicit quantification of EF violations as part of the water boundary, while Gleeson et al. (2020b) subdivided the water planetary boundary into six sub-boundaries and proposed possible control and response variables for each, with aquatic biosphere integrity (i.e., EF) as the potential control variable for a surface water sub-boundary. Quantitative evaluation of the strength and scalability of the identified control and response variables is still required.

## 2. Methodology and Data

The study is conducted at two spatially aggregated scales; 1) global and 2) ecoregion, for a historic time period of 30 years (1976 - 2005). All the underlying calculations were done at level 5 HydroBASIN (median basin area = 19,600 km$^2$) (Lehner and Grill, 2013) and were aggregated to the corresponding spatial scale for further analysis. Level 5 HydroBASIN (also referred to as basin in this paper) was selected as the smallest spatial unit as it is the highest level of specificity that can be rasterized into a 0.5-degree resolution grid without significantly reducing the number of sub-basins smaller than a grid cell (Virkki et al., 2022). The EF violation indices were calculated using Virkki et al. (2022)'s novel Environmental Flow Envelope (EFE) framework, and biodiversity was represented by a combination of relative and absolute value indices. The overall workflow for this manuscript is depicted in Fig. 1.

**2.1 Data**

*2.1.1 Streamflow data*

Streamflow data used in the EFE (see section 2.2 for more details) definition were obtained from the Inter-Sectoral Impact Model Intercomparison Project (ISIMIP) simulation phase 2b outputs of global daily discharge (available at https://esg.pik-potsdam.de) (Warszawski et al., 2014). Monthly streamflow data (averaged from the daily simulations) for two time periods were used in this study; 1) for the pre-industrial era (1800 - 1860), which is considered as the unaltered

reference period (Poff et al., 1997) and 2) for the recent time period (1976 - 2005). These monthly
streamflow datasets were used to calculate EF violations. For calculating the EF violation indices,
the estimated EFEs for each basin were obtained from Virkki et al. (2022). A total of 4 Global
Hydrological Models (GHM) (H08 (Hanasaki et al., 2018), LPJmL (Schaphoff et al., 2018), PCR-
GLOBWB (Sutanudjaja et al., 2018), WaterGAP2 (Müller Schmied et al., 2016)) were used to
obtain the monthly streamflow data. Each GHM was forced with four different Global Circulation
Models (GCM) outputs (GFDL-ESM2M (Dunne et al., 2012), HadGEM2-ES (Collins et al., 2011;
Bellouin et al., 2011), IPSL-CM5A-LR (Dufresne et al., 2013), MICROC5 (Watanabe et al., 2010)).
All the GHM outputs used in this study are extensively validated and evaluated in several previous
studies (e.g., Zaherpour et al., 2018; Gädeke et al., 2020). Moreover, as part of the ISIMIP impact
model intercomparison activity, all the GCM climate input data were bias corrected using
compiled reference datasets covering the entire globe at 0.5 deg resolution (Frieler et al., 2017).
Additionally, the GHM outputs are also validated using historical data to better fit reality (Frieler
et al., 2017). Therefore, no additional volition of the data is done in this study.

The streamflow data were aggregated to the sub-basin scale according to level 5 HydroBASIN
Version 1.0 (https://www.hydrosheds.org/page/hydrobasins) (Lehner and Grill, 2013).  The data
from ISIMIP 2b is representative of historical land use and other human influences including dams
and reservoirs (Frieler et al., 2017). The maximum discharge cell value within the boundaries of
each level 5 HydroBASIN is chosen to represent the outlet discharge value. Any violations within
the outlet cell are regarded as indicative of the entire basin, even if conditions can differ in various
areas within the level 5 HydroBASIN. As the spatial resolution of the study is level 5 HydroBASIN
to allow a global analysis, we accept a certain homogenization of the local scale characteristics.
See supplementary materials (see Supplementary information S.2) for more details on the
datasets used in this study.

*2.1.2 Freshwater biodiversity data*
In addition to the streamflow data, data on fish diversity were also used in this study (Table 1).
Freshwater biodiversity was evaluated using seven indices estimated from the observed biota
data. The biodiversity indicators were obtained from international agencies or the literature. The
biodiversity indicators consisted of six indices of relative change in biodiversity and one index of
absolute values of biodiversity.
*a)  Absolute biodiversity indicator*
The absolute biodiversity indicator consisted of freshwater fish richness (FiR). The fish richness
data was compiled and processed from 1436 published papers, books, grey literature and web-
based sources published between 1960 and 2014 (Tedesco et al., 2017). They cover 3119 basins
all over the world and account for 14953 fish species permanently or occasionally inhabiting
freshwater systems. In addition to FiR, we used the RivFishTIME dataset by Comte et al (2021) –
compiled from long-term riverine fish surveys from 46 regional and national monitoring
programmes and from individual academic research efforts. Though the RivFishTIME dataset is
highly spatially skewed towards the already data rich regions of Europe, North America
(particularly United States of America) and Australia and temporally discontinuous, it is the only
species-specific fish abundance time series data available and is useful to have an independent
verification of the findings using FiR and relative biodiversity indicators.
*b)  Relative biodiversity indicators*
The Relative biodiversity indicators consisted of six freshwater fish facets. Six key facets of
freshwater fish - taxonomic, functional, and phylogenetic diversity (TR, FR, PR respectively), as
well as dissimilarity of each of the three groups (TD, FD, PD respectively)- were used in this
analysis to construct a holistic picture of the state of aquatic biodiversity (see Fig. 1 in Su et al.,
2021 for more details on fish facets calculations). Each facet indicates the change in the
corresponding biodiversity component compared to the 18$^{th}$ century (roughly pre-industrial era).
The taxonomic facets measure the occurrence of fish in a riverine system. Functional facets are
calculated using the morphological characteristics of each species that are linked to feeding and
locomotive functions which in turn relates to larger ecosystem functions like food web control
and nutrition transport. Phylogenetic facets measure the total length of branches linking all
species from the assemblage on the phylogenetic tree. The richness component of the three
categories calculates the diversity among the assemblage whereas the dissimilarity accounts for
the difference between each pair of fish assemblage in one realm. All six fish facets were
calculated at basin scale (2465 river basins) covering 10682 fish species all over the world. The
scale at which the fish facets are estimated, not necessarily align with the scale at which the EF
violations are estimated in all cases. The basin scale facet estimates were then matched with
corresponding EF violation indices using different aggregation/data matching methods (see
section 2.4 for more details). All six facets are available as a single delta change in time and do
not cover multiple timesteps.

Table 1. Details of different data used in this study

| Data | Spatial resolution (extent) | Temporal resolution (extent) | Source/Reference |
|---|---|---|---|
| Aquatic fish richness data | 30 arc second (3119 drainage basins; ~80% of Earth's land) | Temporal aggregate from data compiled from reports between 1960 and 2014 | Observed/Measured data Tedesco et al. (2017) |
| Freshwater fish facets | Basin scale (2465 drainage basins) | Representative of 2015 (change compared to preindustrial era) | Derived from observed data Su et al. (2021) |
| RivFishTIME dataset[4] | Stream reach (11386 sampling location) | 1951 -2019[5] | Comte et al., 2021 |
| EFE | Aggregated to Level 5 HydroBASIN (global) | Monthly (Pre-industrial: 1801-1860) | Model calculated Virkki et al. (2022) |
| Streamflow | Aggregated to Level 5 HydroBASIN (global) | Monthly (Pre-industrial: 1801-1860, Current: 1976-2005) | Model calculated Warszawski et al. (2014) |
| Basin | Level 5 HydroBASIN | Not applicable | Lehner and Grill (2013) |

---

[4] Results only shown in Supplementary Information (see section S8 in Supplementary Information)

[5] Variable for each species and sampling site. Each time-series has a minimum of two-year survey (mean = 8 years).

| boundaries | (global) | | |
|---|---|---|---|


**2.2 Environmental flow violation estimation**

The EFE framework proposed by Virkki et al. (2022) is used to evaluate EF violations in this study. The EFE framework establishes an envelope of variability constrained by discharge limits beyond which flow in the streams may not meet the freshwater biodiversity needs (Virkki et al., 2022). EFE uses pre-industrial (1801-1860) stream discharge to establish an upper and lower boundary for EF deviations at monthly time steps. This EFE is used to define the EF violation at Level 5 HydroBASIN scale. The EF violations were calculated as median ensemble of four Global Hydrological Models (GHM) (H08, LPJmL, PCR-GLOBWB, WaterGAP2) and mean ensemble of four Global Circulation Models (GCM) (GFDL-ESM2M, HadGEM2-ES, IPSL-CM5A-LR, MICROC5). Moreover, five different EF calculation methods (Smakhtin method (Smakhtin et al., 2004), Tennant method (Tennant, 1976), Q90-Q50 (Pastor et al., 2014), Tessmann method (Tessmann, 1979) and Variable Monthly Flow method (Pastor et al., 2014)) were also used in the EFE derivation (see Supplementary Information, Table S3 for more information on EF methods) (Virkki et al., 2022). This approach addresses the uncertainty related to the outputs of models and may eliminate the largest model-related extremes that might cause results to be distorted (Virkki et al., 2022). In spite of the uncertainty in hydrological estimates generated by different models, a simple ensemble matrix often produces acceptable discharge and therefore also EF estimates at larger scales because the bias of the individual models is removed (Zaherpour et al., 2018). Moreover, all the basins with Mean Annual Flow (MAF) < 10 m$^3$/s were excluded due to high uncertainty in EFE and streamflow estimates (Gleeson et al., 2020a; Steffen et al., 2015; Virkki et al., 2022). After this exclusion, a total of 3906 basins were considered for further analysis. However, many low flows are seasonally observed, such that MAF may be quite large due to elevated wet season flows, with extremely low flows during a dry season (e.g., Eel River basin, California) making it difficult to model. In such cases with higher intra annual flow variability, it is appropriate to consider more detailed discharge data (seasonal/sub annual) to gain more insight into the flow modelling uncertainties.


Here we evaluate the EF violation by defining four different EF violation indices: 1) violation
severity (S), violation frequency (F), probability to shift to a violated state (P.shift) and probability
to stay violated (P.stay). Out of the four EF violation indicators, two (S and F) were a modification
from Virkki et al. (2022) and the two (P.shift and P.stay) were calculated based on the current
EFE deviations from Virkki et al. (2022). P.shift and P.stay measures the likelihood of a given year
to shift or stay in a violated state. The state of a basin (violated or non-violated) was identified at
an annual time step and the mean probability to shift or remain in that state is calculated.

The detailed definitions of the EF violation indicators are as follows.

1) Violation severity (S): The annual violation severity was calculated as the absolute mean
of the magnitude of EF deviation from the EFE lower or upper bound in all the violated
months. The magnitude of violation is based on the violation ratio proposed by Virkki et
al. 2022 (See Table S4 in supplementary information). The normalized value of S is used
in this study.
2) Violation frequency (F): Frequency of violation is a measure of the proportion of months
a basin has violated the EFE lower or upper bound in a year. Frequency is calculated as
the percentage of violated months per year. The normalized value of F is used in this
study.
3) Probability to shift to a violated state (P.shift): The P.shift is defined in this paper as the
probability of a basin to shift to a violated state from a non-violated state (Eq. 1). This
indicator along with P.stay gives a measure of the stability of violation in each level 5
HydroBASIN. The violated/non-violated state of a basin is calculated annually based on
the violations in the low flow months. If a basin violates EFE lower or upper bound for at
least three consecutive months during the low flow period (Q<0.4MAF) in a year, then
the basin is considered to be in a violated state.
$$P.shift = \frac{number\ of\ years\ shifted\ to\ violated\ state\ (i.e.year\ i\ is\ violated\ and\ year\ i-1\ is\ not)}{total\ number\ of\ years}$$   (1)

4) Probability to stay violated (P.stay): Once shifted to a violated state, the tendency of a
basin to remain in that state or switch to a non-violated state is determined by this
indicator. If a basin has a higher P.stay (closer to 1) then the basin continues to remain in
the violated state for a longer time before switching to a non-violated state (Eq 2).
Whereas the basins with lower P.stay (closer to 0) tend to remain in the violated state
only for a brief period of time. In other words, the number of consecutive violated years
is much lower for basins with lower P.stay value.
$$P.stay = \frac{number\ of\ violated\ years\ with\ at\ least\ one\ consecutive\ year\ violated}{total\ number\ of\ violated\ years} \qquad (2)$$


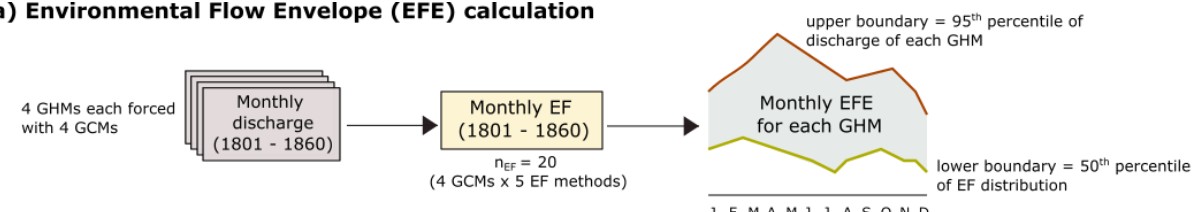

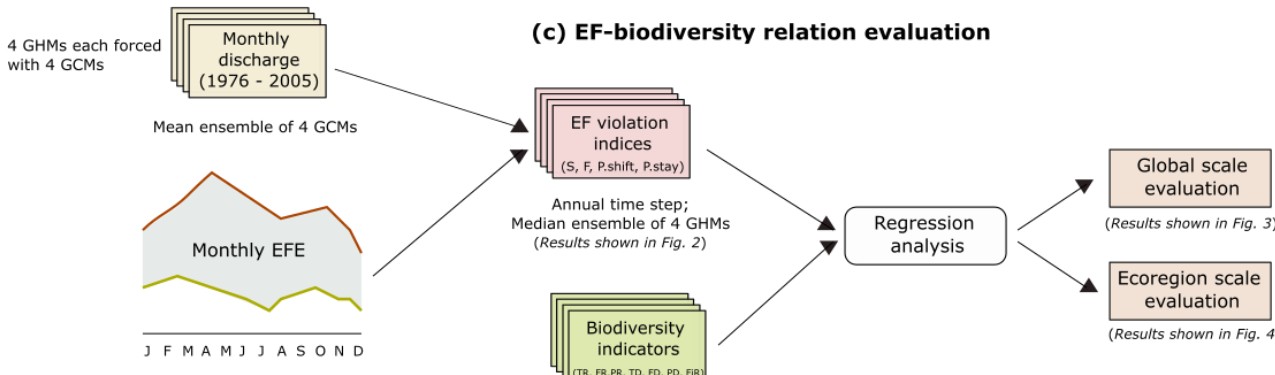


Fig. 1 Methodology outline for (a, b) EF violation indicators calculation and (c)EF-biodiversity
relationship evaluation

**2.3. Relationship between environmental flow violations and freshwater biodiversity**
The relationship between freshwater biodiversity and EF violation was evaluated using regression
analysis. None of the relationships explored in this study exhibited any nonlinearity and hence
first order single variate and multivariate linear regression analysis was opted for this study for
reasons of parsimony and to achieve reasonable correlation accuracy. Further analysis was
carried out by aggregating the level 5 HydroBASIN scale values to global level, WWF's Freshwater
ecoregions major habitat type scale (results given in SI) (Abell et al., 2008) and G200 freshwater
ecoregion level (Olson and Dinerstein, 2002). The G200 freshwater ecoregion is a subset of
WWF's freshwater ecoregion that includes only the biodiversity hotspots. Seven freshwater
ecoregions in ecologically important regions were studied, and the EF-biodiversity relationship
was evaluated separately for each ecoregion type. Aggregating to major ecoregion types
accounts for some data's natural/spatial variability, in addition to using an analysis of global data.

One of the major challenges in conducting an aggregated evaluation was the discrepancy in the
spatial resolution at which the EF violation indices and various biodiversity indicators and the loss
of heterogeneity. Aggregation of any scale will lead to some level of homogenization of the data.
A reach-by-reach evaluation will be an ideal solution to capture all the heterogeneity. However,
this is not very practical for a global study due to data and computational limitations. Therefore,
to partially address this challenge, two different aggregation/data matching methods were
employed; case-1) matching level 5 HydroBASIN data (EF violation indices) to biodiversity data
and case-2) matching biodiversity data to level 5 HydroBASIN (See supplementary information
(SI); Section S5). In the first case every level 5 HydroBASIN (EF violation indices) is matched with
the biodiversity data point nearest centroid. Whereas in the second case there can be three
different scenarios (See SI; Fig. S4): 1) biodiversity basin is smaller than level 5 HydroBASIN; in
that case all the biodiversity basins within one level 5 HydroBASIN were matched with the same
EF violation value, 2) when biodiversity basin is equal in size to level 5 HydroBASIN; in this case
biodiversity basins and level 5 HydroBASIN had a one-to-one match, 3) biodiversity basin is larger
than level 5 HydroBASIN. In the last case, two methods were used for data mapping 1) Outlet
matching: where each biodiversity basin is mapped with EF violation value from the level 5
HydroBASIN closest to the outlet and 2) Mean matching: each biodiversity basin is mapped with
the mean EF violation values of all level 5 HydroBASIN within it. Data matching methods were
employed to partially understand the uncertainty due to scale discrepancy between datasets. As
the results are insensitive to the aggregation method, only the results using case 1 (matching
level 5 HydroBASIN data to biodiversity data) are discussed in this paper.

## 3.Results and Interpretations

### 3.1 Evaluating EF violation drivers and characteristics

The majority of basins face some kind of EF violation (either in terms of severity or frequency or
with higher probabilities to shift and/or stay violated) (Fig. 2). Between 1976 and 2005, 17% and
45% of basins, respectively, experienced violation frequency (F) greater than 3 months/year and
severity (S) greater than 20% from the EFE lower or upper bound (normalized violation index >=
0.25) (Fig.2 a, b). Additionally, 33% of basins have a higher chance of shifting (P.shift >= 0.5; i.e.,
33% basins have over 50% probability to shift to a violated state) to a violated state (Fig.2 c, d).
EF violations are very frequent and severe in mostly arid/semi-arid regions such as the Middle
East, Pakistan, India, Australia, Sahara, Sub-Saharan Africa, Southern Africa, and the
southernmost part of North America. On the other hand, regions with higher probability to shift
to a violated state (P.shift) were not limited to the low precipitation and low streamflow regions.

Although the majority of regions with high P.shift values were arid or semi-arid, some exceptions
included Southeastern Asia and Central South America.  The non-arid regions with higher P.shift
also have extremely high water withdrawal in all sectors (agriculture, domestic and industry).
This spatial concurrence suggests that human activities, as well as hydroclimatic influences, play
a significant role in deciding a region's P.shift. However, once in the violated state, the flow
variability regimes in the catchment determine the probability of remaining (P.stay) in the
violated state. Catchments with highly variable flow regimes (i.e., receive most of the annual flow
as floods; see SI for classification map; Fig. S2) have higher probability to stay violated once
shifted whereas catchments with stable flow regimes (year-round steady high baseflow) have a
higher tendency to revert to a non-violated state. An example of this behavior can be seen in the
Australian basins. Though, almost all the Australian basins have a very high P.shift, only the highly
variable flow regime northern catchments had a higher probability to stay violated. Despite
having an exceedingly high P.shift, the southern stable catchments swiftly shift back to a non-
violated state.

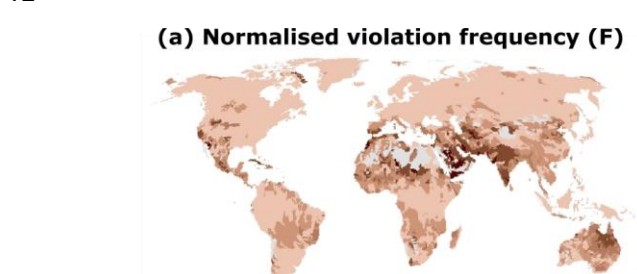
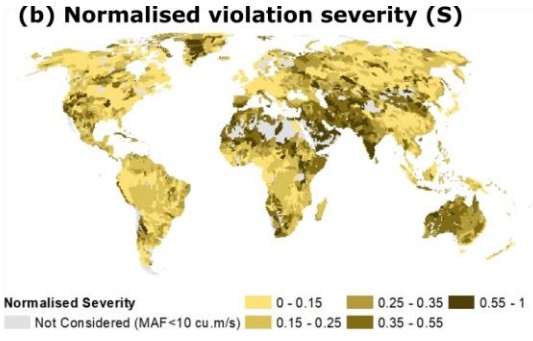
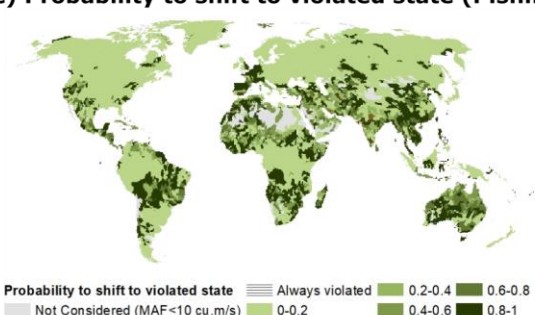
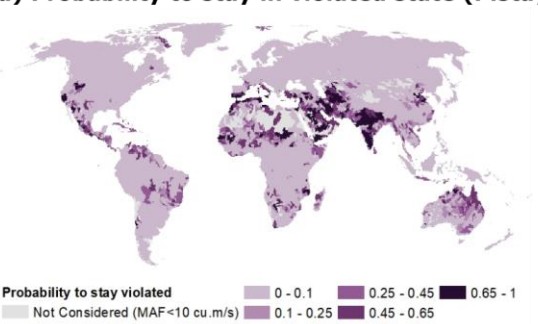


Fig. 2 Four measures of Environmental Flow Envelope (EFE) lower or upper bound violation
estimated using ensemble median of four Global hydrological models; a) Normalized frequency
of violation, b) Normalized severity of violation, c) Probability to shift to a violated state from a
non-violated state and d) Probability to stay violated once shifted to a violated state.

**3.2 Relationship between EF violation and freshwater biodiversity**

The aggregated analysis was carried out at global and ecoregion scales. Multiple aggregation
methods (section 2.3) yielded comparable results, therefore only the case 1 (level 5 HydroBASIN
matched with biodiversity data) results are discussed further (see supplementary material Fig. S5
and S6 for results using other aggregation methods). At the global scale, none of the biodiversity
indicators correlated (significance of p value <0.05) with any EF violation indices (Fig. 2). The
biodiversity indicators were not exhibiting any strong trend in either positive or negative
direction. The correlation coefficient value (R value) for the remaining biodiversity indicators

ranges only from -0.2 to 0.17 (Fig. 3 b). The three fish dissimilarity facets (TD, FD, and PD) show slight negative correlation whereas the richness facets (TR, FR, and PR) display a slight positive correlation with EF violation. The positive correlation of the richness indicators is attributed to an overall increase in the assemblage in most of the basins despite the increase in EF violation. Moreover, (relative) TR and (absolute) FiR were showing opposite trends. The positive trend in TR could be attributed to changes involving nonnative species, whereas the FiR describes the current deteriorated state. The increase in the fish assemblage over time was verified using an independent dataset RivFishTIME (see SI; Fig. S8, Fig. S9) (Comte et al., 2021). The increase in the fish richness facets primarily stems from the introduction of alien species introduced into streams for commercial purposes (Su et al., 2021). The invasion of alien species can tamper with the existing natural ecosystem equilibrium resulting in further degradation of the overall ecosystem health. The results using RivFishTIME data sets were also consistent with the findings using FiR and six relative biodiversity indicators and there was no significant correlation between EF violation indicators and fish abundance data over time (see results for five selected fish species based on data completeness and geographical distribution in Supplementary Information section S8; Fig. S8).

Correlations between EF and biodiversity are generally weak at the scale of G200 freshwater ecoregions as well (see Section 2.2, (Olson and Dinerstein, 2002)). In G200 freshwater ecoregions (see SI; Table S6 for full freshwater ecoregion results) the nature of the EF-biodiversity relationships was highly varying between different ecoregions (Fig 4). In large lakes, large rivers and small lakes, Su et al. (2021) fish richness facets were showing a strong and significant positive correlation with most of the EF violation indices. The increase in biodiversity despite increase in EF violation could be a signal of introduction of nonnative species for commercial purposes. Whereas, in large rivers, large river deltas and xeric basins, the dissimilarity indices, FiR show negative correlation. However, in most ecoregions, the EF-biodiversity relationship is insignificant (p value >0.05). Similar analysis using different aggregation/scale matching methods also yielded comparable results at G200 ecoregion scale (see Fig. S5 and Fig. S6 in Supplementary Information). In addition to this, the multivariate regression analysis results (Fig. 5) also show

very low correlation between EF violation indicators and biodiversity indices in most G200
ecoregion, except in small lakes where the coefficient of determination is between 0.25 - 0.4 for
the richness indicators (TR, FR, PR). The mean coefficient of determination ($r^2$) is approximately
0.1. These results corroborate the above findings that EF violations are not significantly inversely
correlated with biodiversity, regardless of ecoregions with the current dataset.

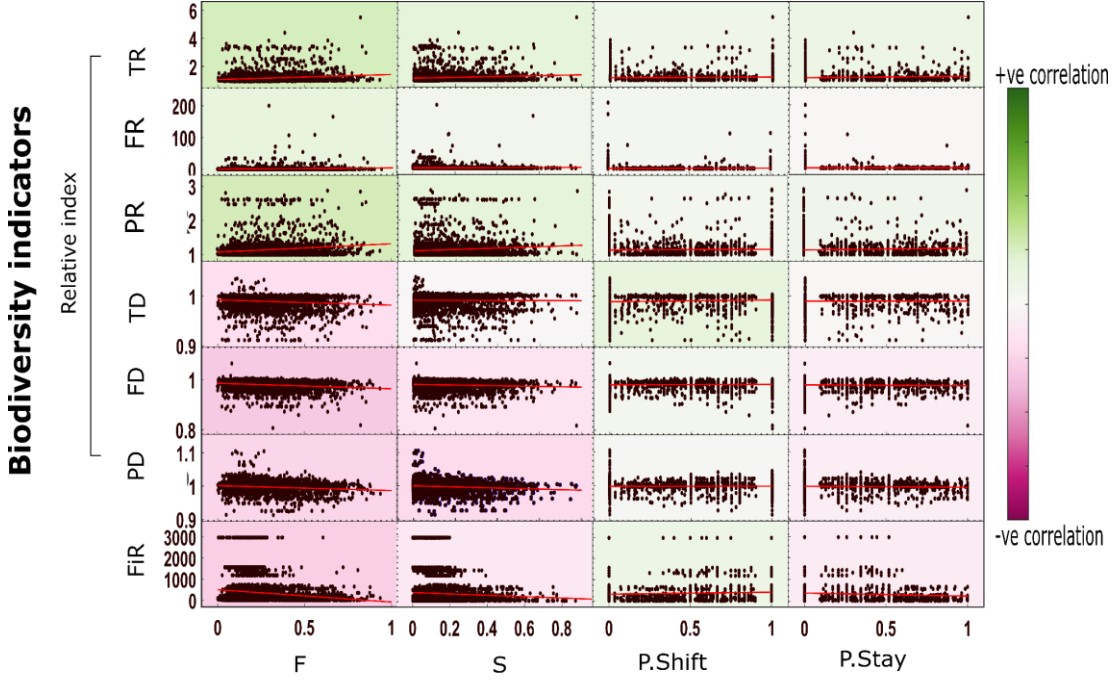

**Environmental Flow violation indicators**
Fig. 3 Scatter between EF violation indices and biodiversity indices with linear fit and
corresponding R value at globally aggregated scale.
Note: This figure represents results from case 1 (level 5 HydroBASIN matched with biodiversity data). The results of
other aggregation methods are given in SI (Fig. S5 and S6).
Abbreviations: F - Frequency of violation; S-Severity of violation; P.shift-Probability to shift to a violated state;
P.stay-Probability to stay in a violated state; FiR-Fish richness; TR-Taxonomic richness; FR-Functional richness; PR-
Phylogenetic richness; TD-Taxonomic dissimilarity; FD-Functional dissimilarity; PD-Phylogenetic dissimilarity

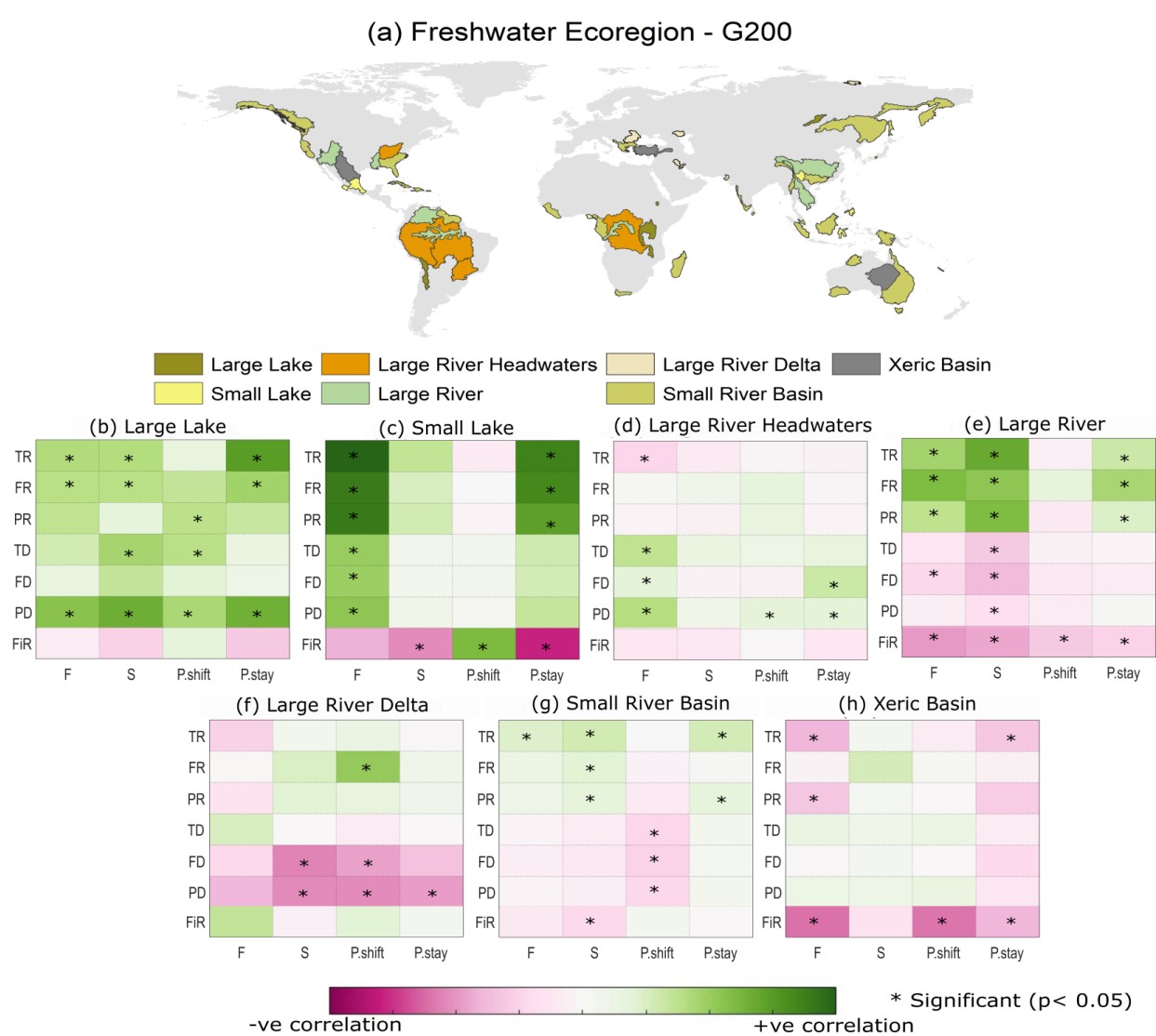

Fig.4 (a) Spatial distribution of different G200 freshwater ecoregions and (b-h) the correlation between EF violation indices and freshwater biodiversity indicators for different G200 freshwater ecoregions.

Note: The results for all the WWF freshwater ecoregions are given in SI (SI section S.7).

Abbreviations: F - Frequency of violation; S-Severity of violation; P.shift-Probability to shift to a violated state; P.stay-Probability to stay in a violated state; FiR-Fish richness; TR-Taxonomic richness; FR-Functional richness; PR-Phylogenetic richness; TD-Taxonomic dissimilarity; FD-Functional dissimilarity; PD-Phylogenetic dissimilarity

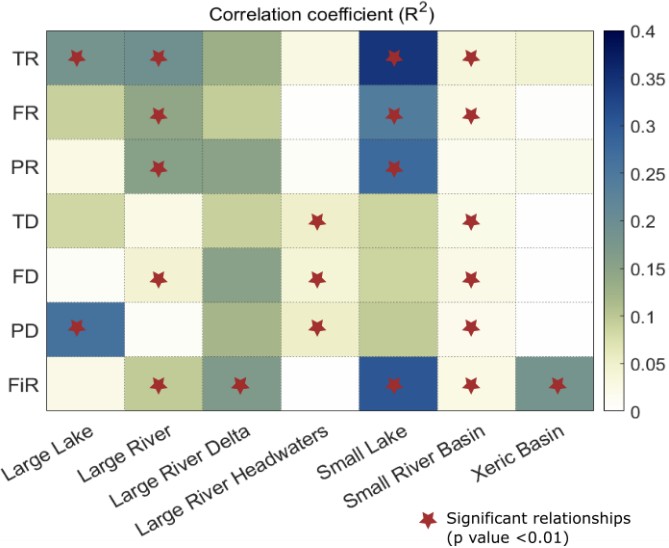

Fig. 5 Coefficient of correlation ($R^2$) for multivariate regression between EF violation indicators
and biodiversity indices. Each row represents on biodiversity indicator and each column
represents one G200 ecoregion

## 4.Discussion

The findings from this study indicate that the EF-biodiversity relationship is poorly correlated at
global or ecoregion scales with currently available data and methods. The most likely explanation
for the lack of correlation is the overwhelming heterogeneity of the freshwater ecosystems - e.g.,
with some freshwater species being more susceptible to variations in flow than others (Poff and
Zimmerman, 2010) - which is not adequately represented in the used spatial resolution (level 5
HydroBASIN). Moreover, when it comes to a larger-scale relationship, several other factors like
climate change (Davies, 2010; Poff et al., 2002), river fragmentation (Grill et al., 2015; Herrera-R
et al., 2020), large-scale habitat degradation (Moyle and Leidy, 1992),  landscaping/river scaping
(Allan et al., 2005), alien species (Leprieur et al., 2008, 2009; Villéger et al., 2011) and water
pollution (Brooks et al., 2016; Shesterin, 2010) can also impact the freshwater ecosystem in
multiple ways. Thus, at Earth system level, other interlinked factors potentially confound the
impact of EF violation on biodiversity degradation.

**4.1 Implications for water management**

The lack of correlation between EF violation and freshwater biodiversity has implications for large-scale water management. A generalized large scale EF approach can underestimate the stress on the ecosystem at a smaller scale where the actual action is taking place. It is undeniable that adequate flow is essential for maintaining freshwater ecosystems. Nonetheless, the current generalized EF estimation methods need further refinement to adequately capture this importance. The global hydrological EF methods are often validated using locally calculated EF requirement values (Pastor et al., 2014) with the assumption of adequate scalability in the EF-biodiversity relationship. However, more holistic EF estimation methods combining hydrological, hydraulic, habitat simulation methods, and expert knowledge (Poff and Zimmerman, 2010; Shafroth et al., 2010) are essential to ensure a healthy freshwater biodiversity. The policies and decisions taken at various scales need a more dynamic framework, where different dominant drivers of ecosystem degradation can be prioritized based on particular cases. For instance, an integrated EF indicator which encompasses quantity, quality, and timeliness of water in the streams will be a better hydrologic indicator to evaluate freshwater ecosystem health than an indicator which accounts only for quantity. Moreover, when making water management decisions, care must be given to account for the temporal and spatial heterogeneity in the ecosystem dynamics.

Although there are some coordinated scientific efforts such as ELOHA (Ecological Limits of Hydrologic Alterations) (Poff et al., 2010) to provide a holistic framework for EF estimation, its scientific complexity and high implementation cost constrains its use around the world (Richter et al., 2012). For example, several European countries like Romania, Czech Republic, Serbia, and Luxembourg use a national level static method to define minimum environmental flows (Linnansaari et al., 2012). Similarly, other jurisdictions use the presumptive standards proposed by Richter et al. (2012) to establish a legal basis for EF protection. These presumptive standards limit hydrologic modifications to a percentage range of natural or historic flow variability. One example of such a case, the North Carolina's Environmental Flow Science Advisory Board uses a presumptive standard of 80-90% of the instantaneous modeled baseline flow as the EF

requirement (NCEFSAB, 2013). The limitation of such a practice is the incorrect presumption of
uniformity in the EF needs over a larger region. Therefore, we recommend the application of
holistic indicators at these large scales (covering all river stretches and tributaries) rather than
using simplified hydrologic-only metrics of EF (violation). However, the authors also acknowledge
the limits in implementation of a more dynamic EF framework in data limited regions. Programs
for more monitoring and data collection and improved, more holistic modeling methods using
more/better data need to be implemented in those regions. Thus, applying a holistic framework
like ELOHA could be made possible and can capture the heterogeneity in the EF-biodiversity
relationship.

**4.2 Implications for a water planetary boundary**
The current rationale in using EF in the water planetary boundary relationship is based on the
assumption of its universal relationship with freshwater biodiversity. However, with the currently
available data and methods the findings for EF-biodiversity relationship are inconclusive.
Moreover, due to the heterogeneity of biodiversity response over time and space, the trend in
any aggregate scale is likely to remain relatively constant instead of showing any discernible
tipping point (Brook et al., 2013). We suggest that to reconsider the use of environmental flows
in defining water planetary boundaries, given the higher degree of heterogeneity and lack of
strength in the ecosystem function-biodiversity relationship. Some of the potential reasons for
the reconsideration are, firstly, freshwater biodiversity may not have pan-regional or
"continental-planetary" scale threshold dynamics, and its link with EF violation might be
inadequate to represent the finer scale variations. Secondly, resource distribution and human
impact heterogeneity suggest the need for regional boundaries as proposed by Steffen et al.
(2015). Thirdly, EF calculation methods used in the current regional/planetary boundary
definition are highly restricted to hydrological methods which may not be adequate to capture
the biodiversity status. A regional boundary transgression can occur even well within planetary-
level safe limits (Brook et al., 2013; Nykvist et al., 2017). Therefore, for an overly complex
biophysical relationship like the EF-biodiversity where multiple shift states are possible, it is
difficult to prioritize and manage critical regions without a regional/local boundary.

**4.3 Limitations and ways forward**


1) **Data scarcity**: Even though this study uses state of the art global hydrological models and best
available global estimates of EF requirements, freshwater ecological data were limited to
freshwater fish. Other than these, several other taxa like crayfish and other benthic
invertebrates, phytoplankton, or zooplankton are also significant in determining the proper
functioning of a freshwater ecosystem (AL-Budeiri, 2021; Domisch et al., 2017; Nyström et al.,
1996). However, due to lack of global data, these taxa are not included in this study. To better
examine the relationship, global datasets for other freshwater biodiversity metrics are urgently
needed.

2) **Discrepancy in data resolution**: The spatial and temporal resolutions at which the EF violation
is estimated here, and the biodiversity indicators measured/calculated are inconsistent. The
basic spatial measuring unit of the biodiversity is sometimes greater or lesser than the basin size
at which EF is measured. This discrepancy could have some impact on the results. However, in
this study several resolution matching methods were used to account for this uncertainty.
Therefore, more detailed data with better-matching scales are needed to overcome this
limitation.

3) **Lack of multi-driver interaction**: In this study, we consider the impact of EF violations on
biodiversity as an independent relationship. In reality, this might not be the case. Other drivers
of ecosystem degradation like land use change, habitat loss, stream modifications and
geographical disconnection can influence the EF-biodiversity relationship. These interactions
were outside the scope of this study but should be taken into account in follow up studies.

4) **Simplified representation of human interference with freshwater systems**: The role of
humans in impairing the ecosystem balance is represented here based on how human water
withdrawals violate hydrologically defined EF. Other human disturbances are thus not accounted
for, such as aquatic habitat degradation through change in land use, artificial introduction of
nonnative species, and non-point pollution from agriculture. Moreover, this study does not
distinguish the climate driven impact on EF violation from the anthropogenic impacts.

5) **Exclusion of impact of dams:** The dams are indeed a large contributing factor to the results
uncertainty. The dam regulated rivers may have a significantly different effect on biodiversity
compared to free-flowing rivers. The ISIMIP data used to calculate EF violations considers the
effects of large dams on streamflow. However, to explicitly isolate the effects of dams in this
analysis from other drivers, the information on dam operation schemes for each sub-basin would
be necessary and this would require a paper on its own. Therefore, the effects of the dams are
incorporated in this study but are not explicitly analyzed separately from other drivers.
**5. Summary and Conclusion**
The relationship between EF violations and freshwater biodiversity is evaluated at globally
aggregated levels in this study. No significant relationship between EF violation and freshwater
biodiversity indicators was found at global or ecoregion scale using globally consistent methods
and currently available data. Relationships may exist at smaller scales and could potentially be
identified with more holistic EF methods including multiple factors (e.g., temperature, water
quality, intermittency, connectivity) and more extensive freshwater biodiversity data. The single
negative result is not a final say but it is a call for conducting more study on existing generalized
and well applied methods.

The paper is not intended to be a definitive test on the relationship between EF and aquatic
biodiversity but more to be an exploratory analysis to tests a widely used but rarely verified
assumption on the relationship at global and ecoregion scale. The lack of correlation in the EF-
biodiversity relationship found in this study suggests taking particular care when developing
macro-scale EF policies (regional and above), and further implies that the conceptualization of a
blue water planetary boundary ought to rest upon a broader set of relationships between
hydrological processes and Earth system functioning. At larger scales, the enormous spatial and
temporal heterogeneity in EF-biodiversity relationship motivates a holistic estimation of EF
grounded on ecosystem dynamics.

## Data Availability

All data to reproduce the analysis in this manuscript is available at
https://borealisdata.ca/dataset.xhtml?persistentId=doi:10.5683/SP3/2BYXZZ and all the code
(Matlab) used is available at https://github.com/ChinchuMohan/Eflows-Biodiversity-Project

## Author Contribution

CM, TG, JSF devised the conceptual and analysis framework of this study with inputs from MK,
MP, and VV. VV performed the EFE calculation with help from MK and MP. CM performed the
biodiversity data compilation and EF-biodiversity analytical evaluation with help from TG, JSF and
XH. CM performed the final analysis and produced the results and visualization shown in the
study, discussing together with TG, JSF, XH, MK, MP, VV and LWE. TG, JSF, MK, MP, VV, LWE, XH,
DG and SCJ contributed to paper writing and the interpretation of the results. CM took the lead
in writing the manuscript. All authors provided critical feedback and helped shape the research,
analysis, and manuscript.

## Compelling Interests

The authors declare no competing interests.

## Acknowledgement

Authors acknowledge various funds that made this research possible. CM received funding from
Canada First Research Excellence Fund (CFRE); MK received funding from Academy of Finland
funded project WATVUL (grant no. 317320), Academy of Finland funded project TREFORM (grant
no. 339834), and European Research Council (ERC) under the European Union's Horizon 2020
research and innovation programme (grant agreement No. 819202). VV received funding from
Aalto University School of Engineering Doctoral Programme and European Research Council
(ERC) under the European Union's Horizon 2020 research and innovation programme (grant
agreement No. 819202). SCJ acknowledges funding through the Leibniz Association for the
project Freshwater Megafauna Futures.

## 567 Supplementary Information

The supplementary information is submitted separately.

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

Declaration, B.: The Brisbane Declaration: environmental flows are essential for freshwater
ecosystem health and human well-being, in 10th International River Symposium, Brisbane,
Australia, 3–6, 2007.
Domisch, S., Portmann, F. T., Kuemmerlen, M., O'Hara, R. B., Johnson, R. K., Davy-Bowker, J.,
Baekken, T., Zamora-Muñoz, C., Sáinz-Bariáin, M., and Bonada, N.: Using streamflow
observations to estimate the impact of hydrological regimes and anthropogenic water use on
European stream macroinvertebrate occurrences, Ecohydrology, 10, e1895, 2017.
Dudgeon, D.: Fisheries: pollution and habitat degradation in tropical Asian rivers, Encyclopaedia
of Global Environmental Change, 3, 2001.
Dudgeon, D.: Prospects for sustaining freshwater biodiversity in the 21st century: linking
ecosystem structure and function, Current Opinion in Environmental Sustainability, 2, 422–430,
643 2010.

Dudgeon, D., Arthington, A. H., Gessner, M. O., Kawabata, Z.-I., Knowler, D. J., Lévêque, C.,
Naiman, R. J., Prieur-Richard, A.-H., Soto, D., and Stiassny, M. L.: Freshwater biodiversity:
importance, threats, status and conservation challenges, Biological reviews, 81, 163–182, 2006.
Dufresne, J.-L., Foujols, M.-A., Denvil, S., Caubel, A., Marti, O., Aumont, O., Balkanski, Y., Bekki,
S., Bellenger, H., and Benshila, R.: Climate change projections using the IPSL-CM5 Earth System
Model: from CMIP3 to CMIP5, Climate dynamics, 40, 2123–2165, 2013.
Dunne, J. P., John, J. G., Adcroft, A. J., Griffies, S. M., Hallberg, R. W., Shevliakova, E., Stouffer, R.
J., Cooke, W., Dunne, K. A., and Harrison, M. J.: GFDL's ESM2 global coupled climate–carbon earth
system models. Part I: Physical formulation and baseline simulation characteristics, Journal of
climate, 25, 6646–6665, 2012.
Frieler, K., Lange, S., Piontek, F., Reyer, C. P., Schewe, J., Warszawski, L., Zhao, F., Chini, L., Denvil,
S., and Emanuel, K.: Assessing the impacts of 1.5 C global warming–simulation protocol of the
Inter-Sectoral Impact Model Intercomparison Project (ISIMIP2b), Geoscientific Model
Development, 10, 4321–4345, 2017.
Gädeke, A., Krysanova, V., Aryal, A., Chang, J., Grillakis, M., Hanasaki, N., Koutroulis, A., Pokhrel,
Y., Satoh, Y., and Schaphoff, S.: Performance evaluation of global hydrological models in six large
Pan-Arctic watersheds, Climatic Change, 163, 1329–1351, 2020.
Gerten, D., Hoff, H., Rockström, J., Jägermeyr, J., Kummu, M., and Pastor, A. V.: Towards a revised
planetary boundary for consumptive freshwater use: role of environmental flow requirements,
Current Opinion in Environmental Sustainability, 5, 551–558, 2013.
Gleeson, T., Wang-Erlandsson, L., Porkka, M., Zipper, S. C., Jaramillo, F., Gerten, D., Fetzer, I.,
Cornell, S. E., Piemontese, L., and Gordon, L. J.: Illuminating water cycle modifications and Earth
system resilience in the Anthropocene, Water Resources Research, 56, 2020a.
Gleeson, T., Wang-Erlandsson, L., Zipper, S. C., Porkka, M., Jaramillo, F., Gerten, D., Fetzer, I.,
Cornell, S. E., Piemontese, L., and Gordon, L. J.: The water planetary boundary: interrogation and
revision, One Earth, 2, 223–234, 2020b.
Gleick, P. H.: Water resources, Encyclopedia of climate, weather, 817–823, 1996.
Gozlan, R. E., Britton, J. R., Cowx, I., and Copp, G. H.: Current knowledge on non-native freshwater
fish introductions, Journal of fish biology, 76, 751–786, 2010.
Grill, G., Lehner, B., Lumsdon, A. E., MacDonald, G. K., Zarfl, C., and Liermann, C. R.: An index-
based framework for assessing patterns and trends in river fragmentation and flow regulation by
global dams at multiple scales, Environmental Research Letters, 10, 015001, 2015.
Hanasaki, N., Yoshikawa, S., Pokhrel, Y., and Kanae, S.: A global hydrological simulation to specify
the sources of water used by humans, Hydrology and Earth System Sciences, 22, 789–817, 2018.
Herrera-R, G. A., Oberdorff, T., Anderson, E. P., Brosse, S., Carvajal-Vallejos, F. M., Frederico, R.
G., Hidalgo, M., Jézéquel, C., Maldonado, M., Maldonado-Ocampo, J. A., Ortega, H., Radinger, J.,
Torrente-Vilara, G., Zuanon, J., and Tedesco, P. A.: The combined effects of climate change and
river fragmentation on the distribution of Andean Amazon fishes, Global Change Biology, 26,
5509–5523, https://doi.org/10.1111/gcb.15285, 2020.
Horne, A. C., Webb, J. A., O'Donnell, E., Arthington, A. H., McClain, M., Bond, N., Acreman, M.,
Hart, B., Stewardson, M. J., and Richter, B.: Research priorities to improve future environmental
water outcomes, Frontiers in Environmental Science, 5, 89, 2017.
Kabat, P., Claussen, M., Dirmeyer, P. A., Gash, J. H., de Guenni, L. B., Meybeck, M., Hutjes, R. W.,
Pielke Sr, R. A., Vorosmarty, C. J., and Lütkemeier, S.: Vegetation, water, humans and the climate:
A new perspective on an interactive system, Springer Science & Business Media, 2004.
Knouft, J. H. and Ficklin, D. L.: The potential impacts of climate change on biodiversity in flowing
freshwater systems, Annual Review of Ecology, Evolution, and Systematics, 48, 111–133, 2017.
Lehner, B. and Grill, G.: Global river hydrography and network routing: baseline data and new
approaches to study the world's large river systems, Hydrological Processes, 27, 2171–2186,
693 2013.

Leprieur, F., Beauchard, O., Blanchet, S., Oberdorff, T., and Brosse, S.: Fish invasions in the world's
river systems: when natural processes are blurred by human activities, PLoS biology, 6, e28, 2008.
Leprieur, F., Brosse, S., Garcia-Berthou, E., Oberdorff, T., Olden, J. D., and Townsend, C. R.:
Scientific uncertainty and the assessment of risks posed by non-native freshwater fishes, Fish and
Fisheries, 10, 88–97, 2009.
Linnansaari, T., Monk, W. A., Baird, D. J., and Curry, R. A.: Review of approaches and methods to
assess Environmental Flows across Canada and internationally, DFO Can. Sci. Advis. Sec. Res. Doc,
701 39, 74, 2012.

Lundberg, J. G., Kottelat, M., Smith, G. R., Stiassny, M. L., and Gill, A. C.: So many fishes, so little
time: an overview of recent ichthyological discovery in continental waters, Annals of the Missouri
Botanical Garden, 26–62, 2000.
Meyer, J. L., Sale, M. J., Mulholland, P. J., and Poff, N. L.: Impacts of climate change on aquatic
ecosystem functioning and health, JAWRA Journal of the American Water Resources Association,
1, 35, 1373–1386, 1999.
Moyle, P. B. and Leidy, R. A.: Loss of biodiversity in aquatic ecosystems: evidence from fish faunas,
in: Conservation biology, Springer, 127–169, 1992.
Müller Schmied, H., Adam, L., Eisner, S., Fink, G., Flörke, M., Kim, H., Oki, T., Portmann, F. T.,
Reinecke, R., and Riedel, C.: Variations of global and continental water balance components as
impacted by climate forcing uncertainty and human water use, Hydrology and Earth System
Sciences, 20, 2877–2898, 2016.
NCEFSAB: Recommendations for estimating flows to maintain ecological integrity in streams and
rivers in North Carolina, 2013.
Nilsson, C., Reidy, C. A., Dynesius, M., and Revenga, C.: Fragmentation and flow regulation of the
world's large river systems, Science, 308, 405–408, 2005.
Nykvist, B., Persson, Å., Moberg, F., Persson, L., Cornell, S., and Rockström, J.: National
environmental performance on planetary boundaries, A study for the Swedish Environmental
Protection Agency (Stockholm Environment Institute, Stockholm), 2017.
Nyström, P. E. R., BRÖNMARK, C., and Graneli, W.: Patterns in benthic food webs: a role for
omnivorous crayfish? Freshwater biology, 36, 631–646, 1996.
Olson, D. M. and Dinerstein, E.: The Global 200: Priority ecoregions for global conservation,
Annals of the Missouri Botanical garden, 199–224, 2002.
Pastor, A. V., Ludwig, F., Biemans, H., Hoff, H., and Kabat, P.: Accounting for environmental flow
requirements in global water assessments, Hydrology and earth system sciences, 18, 5041–5059,
727 2014.

Poff, N. L. and Zimmerman, J. K.: Ecological responses to altered flow regimes: a literature review
to inform the science and management of environmental flows, Freshwater biology, 55, 194–
730 205, 2010.

Poff, N. L., Allan, J. D., Bain, M. B., Karr, J. R., Prestegaard, K. L., Richter, B. D., Sparks, R. E., and
Stromberg, J. C.: The natural flow regime, BioScience, 47, 769–784, 1997.
Poff, N. L., Brinson, M. M., and Day, J. W.: Aquatic ecosystems and global climate change,
Arlington, VA, 44, 1–36, 2002.
Poff, N. L., Richter, B. D., Arthington, A. H., Bunn, S. E., Naiman, R. J., Kendy, E., Acreman, M.,
Apse, C., Bledsoe, B. P., and Freeman, M. C.: The ecological limits of hydrologic alteration
(ELOHA): a new framework for developing regional environmental flow standards, Freshwater
biology, 55, 147–170, 2010.
Poff, N. L., Tharme, R. E., and Arthington, A. H.: Evolution of environmental flows assessment
science, principles, and methodologies, Water for the Environment, Elsevier, 203–236, 2017.
Powell, S. J., Letcher, R. A., and Croke, B. F. W.: Modelling floodplain inundation for
environmental flows: Gwydir wetlands, Australia, Ecological Modelling, 211, 350–362, 2008.
Reid, A. J., Carlson, A. K., Creed, I. F., Eliason, E. J., Gell, P. A., Johnson, P. T., Kidd, K. A.,
MacCormack, T. J., Olden, J. D., and Ormerod, S. J.: Emerging threats and persistent conservation
challenges for freshwater biodiversity, Biological Reviews, 94, 849–873, 2019.
Richter, B., Baumgartner, J., Wigington, R., and Braun, D.: How much water does a river need?,
Freshwater biology, 37, 231–249, 1997.
Richter, B. D., Mathews, R., Harrison, D. L., and Wigington, R.: Ecologically sustainable water
management: managing river flows for ecological integrity, Ecological applications,13, 206–224,
750 2003.

Richter, B. D., Davis, M. M., Apse, C., and Konrad, C.: A presumptive standard for environmental
flow protection, River Research and Applications, 28, 1312–1321, 2012.
Rockström, J., Steffen, W., Noone, K., Persson, Å., Chapin III, F. S., Lambin, E., Lenton, T. M.,
Scheffer, M., Folke, C., and Schellnhuber, H. J.: Planetary boundaries: exploring the safe operating
space for humanity, Ecology and society, 14, 2009.
Schaphoff, S., von Bloh, W., Rammig, A., Thonicke, K., Biemans, H., Forkel, M., Gerten, D., Heinke,
J., Jägermeyr, J., and Knauer, J.: LPJmL4–a dynamic global vegetation model with managed land–
Part 1: Model description, Geoscientific Model Development, 11 (4), 1343–1375, 2018.
Shafroth, P. B., Wilcox, A. C., Lytle, D. A., Hickey, J. T., Andersen, D. C., Beauchamp, V. B.,
Hautzinger, A., McMULLEN, L. E., and Warner, A.: Ecosystem effects of environmental flows:
modelling and experimental floods in a dryland river, Freshwater Biology, 55(1), 68–85, 2010.
Shesterin, I. S.: Water pollution and its impact on fish and aquatic invertebrates, Interactions:
Food, Agriculture And Environment UNESCO Publishing-Eolss Publishers, Oxford, UK, 59–69,
764 2010.

Smakhtin, V., Revenga, C., and Döll, P.: A pilot global assessment of environmental water
requirements and scarcity, Water international, 29 (3), 307–317, 2004.
Smith, M. and Cartin, M.: Water vision to action: catalysing change through the IUCN water and
nature initiative, IUCN, Gland, Switzerland. 2011.
Smith, V. H.: Eutrophication of freshwater and coastal marine ecosystems a global problem,
Environmental Science and Pollution Research, 10 (2), 126–139, 2003.
Steffen, W., Richardson, K., Rockström, J., Cornell, S. E., Fetzer, I., Bennett, E. M., Biggs, R.,
Carpenter, S. R., De Vries, W., and De Wit, C. A.: Planetary boundaries: Guiding human
development on a changing planet, Science, 347(6223), 1259855, 2015.
Su, G., Logez, M., Xu, J., Tao, S., Villéger, S., and Brosse, S.: Human impacts on global freshwater
fish biodiversity, Science, 371, 835–838, 2021.
Sutanudjaja, E. H., Van Beek, R., Wanders, N., Wada, Y., Bosmans, J. H., Drost, N., Van Der Ent, R.
J., De Graaf, I. E., Hoch, J. M., and De Jong, K.: PCR-GLOBWB 2: a 5 arcmin global hydrological and
water resources model, Geoscientific Model Development, 11, 2429–2453, 2018.
Tedesco, P. A., Beauchard, O., Bigorne, R., Blanchet, S., Buisson, L., Conti, L., Cornu, J.-F., Dias, M.
S., Grenouillet, G., and Hugueny, B.: A global database on freshwater fish species occurrence in
drainage basins, Scientific data, 4, 1–6, 2017.
Tennant, D. L.: Instream flow regimens for fish, wildlife, recreation and related environmental
resources, Fisheries, 1, 6–10, 1976.
Tessmann, S. A.: Environmental use sector: reconnaissance elements of the western Dakotas
region of South Dakota study, Water Resources Institute, South Dakota State University, 1979.
Thompson, R. M. and Lake, P. S.: Reconciling theory and practise: the role of stream ecology,
River Research and Applications, 26, 5–14, 2010.
Tickner, D., Opperman, J. J., Abell, R., Acreman, M., Arthington, A. H., Bunn, S. E., Cooke, S. J.,
Dalton, J., Darwall, W., and Edwards, G.: Bending the curve of global freshwater biodiversity loss:
an emergency recovery plan, BioScience, 70(4), 330–342, 2020.
Tonkin, J. D., Olden, J. D., Merritt, D. M., Reynolds, L. V., Rogosch, J. S., and Lytle, D. A.: Designing
flow regimes to support entire river ecosystems, Frontiers in Ecology and the Environment, 19(6),
326–333, 2021.
Tyson, P., Odada, E., Schulze, R., and Vogel, C.: Regional-global change linkages: Southern Africa,
in: Global-regional linkages in the earth system, Springer, 3–73, 2002.
Villéger, S., Blanchet, S., Beauchard, O., Oberdorff, T., & Brosse, S. (2011). Homogenization
patterns of the world's freshwater fish faunas. Proceedings of the National Academy of Sciences,
798    108(44), 18003-18008.

Virkki, V., Alanärä, E., Porkka, M., Ahopelto, L., Gleeson, T., Mohan, C., Wang-Erlandsson, L., Flörke, M.,
Gerten, D., Gosling, S.N. and Hanasaki, N.: Environmental flow envelopes: quantifying global,
ecosystem–threatening streamflow alterations, Hydrology and Earth System Sciences,1–31,
Vitousek, P. M., Mooney, H. A., Lubchenco, J., and Melillo, J. M.: Human domination of Earth's
ecosystems, Science, 277(5325), 494–499, 1997.
Vitule, J. R. S., Freire, C. A., and Simberloff, D.: Introduction of non-native freshwater fish can
certainly be bad, Fish and fisheries, 10(1), 98–108, 2009.
Vörösmarty, C. J., Wasson, R., and Richey, J. E.: Modelling the transport and transformation of
terrestrial materials to freshwater and coastal ecosystems workshop report, International
Geosphere Biosphere Programme [Stockholm], 1997.
Vorosmarty, C. J., Green, P., Salisbury, J., and Lammers, R. B.: Global water resources:
vulnerability from climate change and population growth, Science, 289(5477), 284–288, 2000.
Vörösmarty, C. J., McIntyre, P. B., Gessner, M. O., Dudgeon, D., Prusevich, A., Green, P., Glidden,
S., Bunn, S. E., Sullivan, C. A., and Liermann, C. R.: Global threats to human water security and
river biodiversity, Nature, 467(7315), 555–561, 2010.
Warszawski, L., Frieler, K., Huber, V., Piontek, F., Serdeczny, O., and Schewe, J.: The inter-sectoral
impact model intercomparison project (ISI–MIP): project framework, Proceedings of the National
Academy of Sciences, 111(9), 3228–3232, 2014.
Watanabe, M., Suzuki, T., O'ishi, R., Komuro, Y., Watanabe, S., Emori, S., Takemura, T., Chikira,
M., Ogura, T., and Sekiguchi, M.: Improved climate simulation by MIROC5: mean states,
variability, and climate sensitivity, Journal of Climate, 23(23), 6312–6335, 2010.
Wilting, H. C., Schipper, A. M., Bakkenes, M., Meijer, J. R., and Huijbregts, M. A.: Quantifying
biodiversity losses due to human consumption: a global-scale footprint analysis, Environmental
Science & Technology, 51(16), 3298–3306, 2017.
Xenopoulos, M. A., Lodge, D. M., Alcamo, J., Märker, M., Schulze, K., and Van Vuuren, D. P.:
Scenarios of freshwater fish extinctions from climate change and water withdrawal, Global
change biology, 11(10), 1557–1564, 2005.
Yoshikawa, S., Yanagawa, A., Iwasaki, Y., Sui, P., Koirala, S., Hirano, K., Khajuria, A., Mahendran,
R., Hirabayashi, Y., and Yoshimura, C.: Illustrating a new global-scale approach to estimating
potential reduction in fish species richness due to flow alteration, Hydrology and Earth System
Sciences, 18(2), 621–630, 2014.
Zaherpour, J., Gosling, S. N., Mount, N., Schmied, H. M., Veldkamp, T. I., Dankers, R., Eisner, S.,
Gerten, D., Gudmundsson, L., and Haddeland, I.: Worldwide evaluation of mean and extreme
runoff from six global-scale hydrological models that account for human impacts, Environmental
Research Letters, 13(6), 065015, 2018.