# Peer review of "Poor correlation between large-scale environmental flow"

_Hydrology and Earth System Sciences, 2022_

## Author Comment (AC1)

**Response to Anonymous Referee #1's comments on manuscript hess-2022-87**

In this document, authors respond to comments from Referee 1 (response to referee 2 and community comment added as additional information at the end of the document for reference) regarding the manuscript titled, "Poor correlation between large-scale environmental flow violations and freshwater biodiversity: implications for water resource management and the water planetary boundary.".

Authors sincerely thank Anonymous Referee #1 for their constructive comments which have helped to improve the article. We address each comment in turn below.

Note: All line numbers in responses correspond to revised manuscript.

**Referee 1**

Comment 1.1: EF violation does not explain changes in biodiversity at the scales of consideration. It would seem that the authors have the tools and the datasets to address some of the major challenges they discuss. In particular, the authors recognize scale and scale matching as issues, and even discuss a solution for dealing with it (Line 266).

"*Aggregation of any scale will lead to some level of homogenization of the data. A reach-by-reach evaluation will be an ideal solution to capture all the heterogeneity. However, this is not very practical for a global study due to data and computational limitations.*"

I would ask the authors: Why is this analysis impractical? None of the metrics being calculated in the manuscript are computationally complex, and the most complex statistical technique is to regress change in biodiversity onto change in EFmetric. Importantly, the authors list scale-matching as a potential explanation for their null finding (Line 445). Of all the limitations listed, this one seems the most straightforward to address in the present manuscript without needing to find new datasets or formulate a more complex model (e.g. reasons 1, 3, 4, 5 in Section 4.3). There are likely other simple checks the authors could perform to explore whether the results might change if the scale-matching is performed differently. For example, how strongly do EF violations correlate within a watershed? Alternatively, if EF violation is observed at the outlet, do we see that an elevated fraction of sub-watersheds also exhibit EF violation (maybe test using some form of logistic regression)? Or, can the authors show that the results are more robust in the watersheds where scales ARE matched (e.g. Figure S4b)? These may help us to answer whether we might expect the results to change if the analysis was performed at a different (e.g. reach) scale, or if scales were more appropriately matched.

Response 1.1: As discussed in the revised manuscript (Line 294 -295), a reach by reach or a finer resolution comparison of the EF violation and biodiversity indicators might be an ideal way to capture the heterogeneity of the aquatic ecosystems. However, to our understanding, there are no fine resolution gridded datasets available for biodiversity except of the ones derived based on

streamflow deviations (Mean Species Abundance from GLOBIO-Aquatic, [*Janse et al.*, 2015]). Comparing two model derived values rooted on streamflow deviations would not satisfy the overarching aim of this study, especially when both are based on same underlying process assumptions, would not add insight into biodiversity responses to EF violations and hence is not included in the manuscript.

Additionally, to partially address the heterogeneity limitation, in addition to our global study, we used the RivFishTIME dataset by Comte et al (2021) – compiled from long-term riverine fish surveys from 46 regional and national monitoring programmes and from individual academic research efforts – and repeated the analysis with the proposed EF violation indicators.

The results were consistent with the findings of the main study and there was no significant correlation between EF violation indicators and fish abundance data over time (see results for five selected fish species based on data completeness and geographical distribution in Supplementary Information section S8). The details of the RivFishTIME dataset and the findings using this dataset are also included in the revised manuscript (see Table 1, Line 180 - 186 and Line 359 – 368)

Line 180 -186: In addition to FiR, we used the RivFishTIME dataset by Comte et al (2021) – compiled from long-term riverine fish surveys from 46 regional and national monitoring programmes and from individual academic research efforts. Though the RivFishTIME dataset is highly spatially skewed towards the already data rich regions of Europe, North America (particularly United States of America) and Australia and temporally discontinuous, it is the only species-specific fish abundance time series data available and is useful to have a independent verification of the findings using FiR and relative biodiversity indicators.

Line 359-368: The increase in the fish assemblage over time was verified using an independent dataset RivFishTIME (see SI; Fig. S8, Fig. S9) (Comte et al., 2021). The increase in the fish richness facets primarily stems from the introduction of alien species introduced into streams for commercial purposes (Su et al., 2021). The invasion of alien species can tamper with the existing natural ecosystem equilibrium resulting in further degradation of the overall ecosystem health. The results using RivFishTIME data sets were also consistent with the findings using FiR and 6 relative biodiversity indicators and there were no significant correlation between EF violation indicators and fish abundance data over time (see results for five selected fish species based on data completeness and geographical distribution in Supplementary Information section S8; Fig. S8).

With respect to the scale matching, analysis similar to the one shown in main manuscript are carried out using different aggregation/scale matching techniques and the results are included in the Supplementary Information (see Fig. S5 and S6). Additionally, the manuscript is revised to reflect the results from different scale matching techniques (see Line 295-299 and Line 379-380)

Additionally, as suggested by the Referee, when revising the manuscript we estimated the variance of EF violation indicators within the catchment boundary (consistent with Su et al.'s

facets). The results were added in the Supplementary material (See section S10, Fig. S12 in Supplementary Information).

[Figure]

(a) Frequency of violation          (b) Severity of violation

Coefficient of variation

0                                    2

Fig. S12 EF violation indicators' coefficient of variance within fish facets data catchment boundary (Su et al., 2021): EF violation (a) frequency and (b) severity

Comment 1.2: There may be some confusion for "uninitiated" readers regarding terminology in the abstract. I suggest defining important terms, like "EF violation", "a planetary boundary for freshwater", etc. I was unfamiliar with some of these terms

Response 1.2: Definitions of uncommon terms are provided in footnotes along with the abstract.

Comment 1.3: Line 214: The authors exclude catchments with MAF < 10 cms. However, many low flows are seasonally observed, such that MAF may be quite large due to elevated wet season flows, with very low flows during a dry season. This is definitely the case in California. Even though many Coastal CA Level 5 watersheds have MAF > 10 cms, low flows during the dry season can be very small, and difficult to model (e.g. the Eel River, Level 5 basin 7050014040). Yet, many of the most important EF metrics are based on low flows, as these represent the period of the year when water is most limiting for ecosystems. In general, I'd like to hear a bit more about the success of the ISIMIP flow model in these seasonally dry low flow watersheds, given how finicky many low flow EF metrics can be. Section 2.1: In general, given the challenges associated with hydrological models, have the authors considered using the gage data used to cal/val ISIMIP? Understandably there will be no pre-industrial record, but presumably trends in EF metrics could be calculated in "early" (e.g. 1980) versus "late" (e.g. 2015) periods, and any associated trend in biodiversity metric could be explored? This would circumvent any issues with low flow modeling.

Response 1.3: All the GHM outputs used in this study are extensively validated and evaluated in several previous studies (e.g. Gädeke et al., 2020; Zaherpour et al., 2018). Moreover, as part of the ISIMIP impact model intercomparison activity, all the GCM climate input data were bias corrected using compiled reference datasets covering the entire globe at 0.5 deg resolution (Frieler et al., 2017). Additionally, the GHM outputs are also validated using historical data to better fit reality (Frieler et al., 2017). Though seasonal performance of ISIMIP data is not conducted at global scale, there are several studies that evaluates the seasonal performance of GHMs at large basin scales (Huang et al., 2016; Gädeke et al., 2020; Zaherpour et al., 2018). All

these studies report reasonable performance in capturing the seasonal dynamics by the GHMs. We thus think that performing a global-scale validation of discharge is not required again, and beyond the scope of an application study like this (see lines 151 - 156). The authors, however, agree that the current analysis was carried out at annual time step which overlooks the seasonal variations in the EF-biodiversity relationships. In the revised manuscript, we will take note of this special case where the intra annual variability in discharge is very high and implicate that higher detail - both in sub-basin catchment boundaries and sub annual discharge data - would be required for practical evaluation uncertainty these cases (see lines 230-234)

Comment 1.4: Line 191 Are the biogeographical realms just the base spatial units of the biodiversity datasets? E.g. the gray shapes in Figure S4b?

Related to the previous question, I'm having some trouble understanding some of the biodiversity metrics, and how they relate to scale. This is probably just some confusion on terminology on my part; I'm a bit new to this particular topic.

So, for example, can the authors more clearly define "dissimilarity" (Line 181)? It is stated that it, "...accounts for the difference between each pair of fish assemblage in one biogeographical realm." It would be really helpful to have some basic equations here, and some explanation of how the calculations correspond to the different scales of aggregation discussed in the flow section and the aggregation section.

Response 1.4:  Biogeographical realms (ecoregions) are the spatial units used in this study for classifying the results into hydro-ecologically similar groups. The grey shapes in Fig. S4b are the spatial scale of relative freshwater fish facets (TR, FR, PR, TD, FD, PD) obtained from Su et al. 2021. The freshwater fish richness data (FiR), from Tedesco et al., 2017, however is at 30 arc second and is restricted to 3119 drainage basins. The spatial and temporal scale of individual data is included in Table S1 in Supplementary Information.

In order to better visually explain the concept of richness and dissimilarity fish facets, a reference to figure 1 in Su et al., 2021 is added in the manuscript (Lines 201-205). Additionally, the following sentence explaining the how the calculations correspond to the different scales of aggregation

Line 201-205: "The scale at which the fish facets are estimated does not necessarily align with the scale at which the EF violations are estimated in all cases. The basin scale facet estimates were then matched with corresponding EF violation indices using different aggregation/data matching methods (see Section 2.4 for more details)."

Comment 1.5: line 231 - First it is stated this is calculated as the absolute mean of the deviation magnitude, but then it is normalized? Should this be interpreted as a percent change in the mean? Is this how the other metrics (e.g. F) are also normalized?

Response 1.5: Annual violation severity (S) was calculated by taking the mean of the magnitude of monthly deviation beyond upper or lower EFE boundary. The magnitude of violation is based on the violation ratio proposed by Virkki et al. 2022.

Table 2 from Virkki et al., 2022. Computing the EFE violation ratio. Q stands for monthly discharge between 1976 and 2005; $EFE_{lower}$ for the EFE lower bound, and $EFE_{upper}$ for the EFE upper bound

| Condition | Violation ratio equation | Violation ratio value |
|---|---|---|
| $Q < EFE_{lower}$ | $\dfrac{Q - EFE_{lower}}{EFE_{lower}} x100$ | < 0 |
| $EFE_{lower} \leq Q \leq EFE_{upper}$ | $\dfrac{Q - EFE_{lower}}{EFE_{upper} - EFE_{lower}} x100$ | 0 -100 (no violation) |
| $Q > EFE_{upper}$ | $(\dfrac{Q - EFE_{upper}}{EFE_{upper}} + 1 )x100$ | >100 |

According to this definition, the lower bound violations will have negative magnitude while the upper bound will have positive magnitude. Therefore, the absolute values were taken in this study to avoid mutual cancellation of the upper and lower bound violations. Additionally, in order to make the different EF violation indices comparable, the values of violation indicators (F and S) were scaled (or normalized) between 0 to 1 using the following formula

$$X_{normal} = \frac{X - X_{min}}{X_{max} - X_{min}}$$

where, $X_{normal}$ = normalised value; X = actual value; $X_{min}$= minimum value in the dataset; $X_{max}$ = maximum value in the dataset

Comment 1.6: Line 236: On the probability of a shift from nonviolated to violated. Is total years in the denominator incorrect? I would think it would be conditional on the occurrence of a nonviolated state, as you can't shift from nonviolated to violated if you're currently in a violated state.

Response 1.6: The logic behind using the total number of years in the denominator was to estimate the probability to shift given the entire period of time. This enabled easy and logical comparison between different regions. Authors, however, agree with the Referee that using non violated years in the denominator is also an alternate way of looking at this shift.

**Response to Referee 2 and Community comment for additional reference**

**Referee 2**

Comment 2.1: This manuscript is mostly about correlation analysis. However, it is not clear to me from the manuscript what correlation analysis method was used by the authors and justification was not provided.

Response 2.1: None of the datasets used in this study exhibited nonlinearity. Therefore, this study uses first order linear regression analysis to evaluate the EF-biodiversity relationship. Additionally considering the suggestion from Referee 2, a multivariate regression analysis was also carried out to evaluate the combined impact of different EF violation indices.

An explanation on the choice of regression analysis is added in the methodology section of the paper.

Line 278 - 281: 'The relationship between freshwater biodiversity and EF violation was evaluated using regression analysis. None of the relationships explored in this study exhibited any nonlinearity and hence first order single variate and multivariate linear regression analysis was opted for this study for reasons of parsimony and to achieve reasonable correlation accuracy. '

Comment 2.2: A second comment is on the use of correlation only. Why look at this on a one vs. one variable basis? Why not develop appropriate statistical approaches to look into the effects of the explanatory variables at the same time. which can also provide statistical significance?

Response 2.2: Considering the suggestion from the Referee 2, a multivariate regression analysis was carried out for each G200 ecoregion when revising the paper. These new results were added in the manuscript (See section 3.2, Fig. 5). The results are in line with the single variable linear regression analysis given in the main manuscript. The mean coefficient of determination ($r^2$) is approximately 0.1.

The following explanation on the multivariate regression is also added in the manuscript

Line 380-384: In addition to this, the multivariate regression analysis results (Fig. 5) also show very low correlation between EF violation indicators and biodiversity indices in most G200 ecoregion, except in small lakes where the coefficient of determination is between 0.25 - 0.4 for the richness indicators (TR, FR, PR). The mean coefficient of determination ($r^2$) is approximately 0.1.

[Figure]

Fig. 5 Coefficient of correlation (r2) for multivariate regression. Each row represents on biodiversity indicator and each column represents one G200 ecoregion

Comment 2.3: I am not sure if Box 1 is needed or if it follows the HESS journal guidelines. Why not just provide these paragraphs in the manuscript text?

Response 2.3: Adding the information in Box 1 could disrupt the overall flow of the manuscript. The information in Box 1 could however aid the readers who are not very familiar with the planetary boundary concept or who want to know more about it. The HESS manuscript format mandates do not explicitly say that boxes are not allowed.

Comment 2.4: Data: While Table 1 provides a nice summary of the various data in this study, a flowchart diagram is strongly recommended to help readers to understand the different underlying layers, e.g., the different variables, the different EF calculation methods, the different GCM models, etc.

Response 2.4: A flowchart summarizing the EF violation indicators calculation is added (see new Fig. 1)

[Figure]

[Figure]

Fig. 1 Methodology outline for (a,b) EF violation indicators calculation and (c)EF-biodiversity relationship evaluation

Comment 2.5: Data: For S and F, the authors says that these variables are normalized. Please be more clear on the normalization.

Response 2.5: Inorder to make the different EF violation indices comparable, the values of violation indicators (F and S) were scaled (or normalised) between 0 to 1 using the following formula

$$X_{normal} = \frac{X - X_{min}}{X_{max} - X_{min}}$$

where, $X_{normal}$ = normalised value; X = actual value; $X_{min}$= minimum value in the dataset; $X_{max}$ = maximum value in the dataset.

Please refer to Response 1.5 for more information

Comment 2.6: Line 294: Is it redundant to list Middle East, Iran, and Iraq?

Response 2.6: Thanks for pointing this out. Iran and Iraq are removed as those are indeed part of Middle East (Line 319-321)

Comment 2.7: Line 327: I don't think "negative trend" is the right word choice.

Response 2.7: We agree. The word negative trend is replaced with negative correlation (Line 353 and Line 377).

**Community Comment 1 - Davy Vanham**

Comment 3.1: Here they put as aim (lines 107-108) " *In order to scientifically underpin large scale EF policies, the existing assumption of the inverse relationship between freshwater biodiversity response and EF violation must be tested regional and global scales*"

However, in this paper I see a mismatch between the analysis conducted on the one hand and the interpretation/discussion on the other hand.

The authors only look at environmental flows. They acknowledge that the violation of environmental flows influences aquatic biodiversity (eg lines 84-86). As they discuss in lines 368-374, other factors influence aquatic biodiversity: climate change, river fragmentation, large-scale habitat degradation, landscaping/river scaping, alien species introduction and water pollution. They however do not account for these factors in their analysis.

Response 3.1: The primary motivation for conducting this study is due to the fact that, majority of the methods used to estimate EF operate at a coarser (global) scale with an underlying assumption that the proportion of flow allocated directly impacts the ecosystem health. However, what this study is doing is reevaluating this assumption. As raised in the discussion of this paper, a holistic approach including bio-geo-hydro-physical approach is necessary to ensure proper functioning of associated ecosystems. Authors agree with the commenter that it is necessary to evaluate the influence of non-hydrologic factors on aquatic ecosystem wellbeing. Moreover, the climate change impact is indirectly taken into account in the EFE analysis (Virkki et al., 2022). Moreover, we have also included the aspect of other confounding environmental factors that might strongly influence the result into the discussion/limitation section (see lines 500-504)

Comment 3.2: I do not follow the reasoning in section 41. It is clear that there are many different EF methodologies, and that more holistic EF estimation methods are required for water management. But this has been expressed by many (recent studies), such as https://doi.org/10.1016/S2542-5196(21)00234-5 or many others. It is not the authors of this study who prove that with their analysis. I doubt whether for more holistic EF methods better correlations can be found, as long as the many other factors are not taken into account.

Response 3.2: By 'holistic approach', authors mean the inclusion of non-hydrologic factors. Through this study, we try to promote a more inclusive approach in estimating the flow requirements for freshwater ecosystems. This idea is supported in this paper by quantitatively evaluating whether environmental flow is the only or key driver of aquatic biodiversity. Authors are in complete agreement with the commenter that a holistic approach which is not limited to the quantity of water in the streams is a better alternative to conventional EF methodologies. This message is emphasized throughout the entire paper acknowledging the literature before.

Comment 3.3: I also do not see why this analysis has implications for a water planetary boundary (section 42). What you only show is that global assessments, due to data restrictions and assumptions, lead to quite some uncertainty. But that does not mean the current bottom up methodology using EFs would be lacking, it just means the boundary has

a wide uncertainty range. EF do provide a meaningful boundary for freshwater biodiversity. That is why it is used in SDG indicator 642, a very significant upgrade from the millennium goal on water scarcity. You actually have a methodology that has global monitoring obligations for UN member states, thereby making it directly policy relevant. Due to the fact that you do not account for the other factors affecting aquatic biodiversity, and therefore do NOT prove inconsistency in "…universal relationship with freshwater biodiversity" (line 416-418) I do not see any justification for the statement "We suggest that to reconsider the use of environmental flows in defining water planetary boundaries" (line 421-422).

Response 3.3: There are several studies proposing environmental flow transgressions as a potential control variable for defining the safe operating space for a freshwater planetary boundary (Steffen et al., 2015; Gerten et al., 2013). However, these assumed relationships between streamflow and aquatic biodiversity have not been studied at global or large regional scales. Therefore, as mentioned in the previous response, this study aids in testing a widely used but unverified assumption on the relationship between environmental flow and aquatic biodiversity at global and ecoregion scale.

Comment 3.4: To conclude, I recommend that the authors re-evaluate their section 4, as well as conclusions, abstract and title. As an example, for the key research points (with in capital letters recommendations):
- No significant relationship between environmental flow (EF) violation and freshwater biodiversity indicators was found at global or ecoregion scales using globally consistent methods and currently available data, WHEN NOT ACCOUNTING FOR OTHER FACTORS AFFECTING FRESHWATER BIODIVERSITY
- Several basins show a slight positive correlation between EF violation and biodiversity indicators, which could be attributed to the artificial introduction of non-native species. HOW IS THE INFLUENCE OF FACTOR NON-NATIVE SPECIES PROVEN? WHAT WITH THE OTHER FACTORS?
- A generalized approach that incorporates EF considerations but ignores the lack of a significant EF-biodiversity relationship at large scales can underestimate the stress on the ecosystem at smaller scales which correspond with eco-hydrological processes that determine ecological impacts from EF violation. NOT CLEAR, AS YOU DO NOT ACCOUNT FOR OTHER FACTORS. ALSO, THESE OTHER FACTORS ARE ESSENTIAL FROM LOCAL TO GLOBAL SCALE, THEY WILL DETERMINE AT ALL SCALES THE CORRELATION between environmental flow (EF) violation and freshwater biodiversity. WHAT YOU PROBABLY MEAN IS WHAT GLOBAL MODELS ARE ABLE TO CAPTURE. But then, future data availability will only improve making multi-regression assessments

Response 3.4: Necessary changes are made in the manuscript

Comment 3.5: Lines 363-374: again, poor correlation by ignoring these other factors. Lines 364-368: no, the other factors are determining. Line 368: no, not only for larger-scale relations, also on a local level. The sudden introduction of a point source pollution can plumet aquatic biodiversity on a very small scale, and therefore also on this small scale, even with very detailed data availability, the other factors need to be accounted for when looking at correlations

Response 3.5: Please refer to response 3.2

Comment 3.6: Ps I also think that putting a title like "Poor correlation between large-scale environmental flow violations and freshwater biodiversity", is not helpful for implementing EFs in the field or policy agendas. As said, its inclusion in SDG indicator 642 is a major advancement and international success. Your title could be misused for not acting on preserving or rehabilitating EFs. When not put in context, some could use it as a slogan not to act on EFs.

Response 3.6: The paper is not intended to be a definitive test to disprove the relationship between EF and aquatic biodiversity. It is intended to be an exploratory analysis to identify the validity of the relation. We do not, in any way, intend to disregard the importance of flow,  but instead our aim is to estimate the usability of large scale generalized EF estimation methods by evaluating its relationship to aquatic biodiversity indicators. The single negative result is not a final say but it is a call for conducting more study on existing generalized and well applied methods.

We acknowledge the risk of reporting a non-correlation between EF and biodiversity. To avoid the risk of misjudgment by the readers, we have strengthened the discussion that our findings are only applicable at global or ecoregion scale and with currently available data. At a scale smaller than this, several studies have already proved the importance of flow for maintaining ecosystem services. The authors, however, think it is more appropriate to keep the title unchanged to be upfront, simple and honest about the findings.

Necessary changes are made in the abstract and conclusion section to minimize the chances of miscommunication of the intended purpose of the paper.

**Reference**

Comte, L., J. Carvajal-Quintero, P. A. Tedesco, X. Giam, U. Brose, T. Erős, A. F. Filipe, M. J. Fortin, K. Irving, and C. Jacquet (2021), RivFishTIME: A global database of fish time-series to study global change ecology in riverine systems, *Global Ecology and Biogeography*, *30*(1), 38-50.

Gerten, D., H. Hoff, J. Rockström, J. Jägermeyr, M. Kummu, and A. V. Pastor (2013), Towards a revised planetary boundary for consumptive freshwater use: role of environmental flow requirements, Current Opinion in Environmental Sustainability, 5(6), 551-558.

Huang, S., Kumar, R., Flörke, M., Yang, T., Hundecha, Y., Kraft, P., Gao, C., Gelfan, A., Liersch, S., Lobanova, A., Strauch, M., van Ogtrop, F., Reinhardt, J., Haberlandt, U., and Krysanova, V.: Evaluation of an ensemble of regional hydrological models in 12 large-scale river basins worldwide, Climatic Change, 141, 381–397, https://doi.org/10.1007/s10584-016-1841-8, 2017.

Janse, J., J. Kuiper, M. Weijters, E. Westerbeek, M. Jeuken, M. Bakkenes, R. Alkemade, W. Mooij, and J. Verhoeven (2015), GLOBIO-Aquatic, a global model of human impact on the biodiversity of inland aquatic ecosystems, *Environmental Science & Policy*, *48*, 99-114.

Steffen, W., K. Richardson, J. Rockström, S. E. Cornell, I. Fetzer, E. M. Bennett, R. Biggs, S. R. Carpenter, W. De Vries, and C. A. De Wit (2015), Planetary boundaries: Guiding human development on a changing planet, *Science*, *347*(6223).